# MicroRNAs and Apoptosis in Colorectal Cancer

**DOI:** 10.3390/ijms21155353

**Published:** 2020-07-28

**Authors:** Hsiuying Wang

**Affiliations:** Institute of Statistics, National Chiao Tung University, Hsinchu 30010, Taiwan; wang@stat.nctu.edu.tw

**Keywords:** apoptosis, colorectal cancer, microRNA, resistance

## Abstract

Colorectal cancer (CRC) is the third leading cause of cancer death in the world, and its incidence is rising in developing countries. Treatment with 5-Fluorouracil (5-FU) is known to improve survival in CRC patients. Most anti-cancer therapies trigger apoptosis induction to eliminate malignant cells. However, de-regulated apoptotic signaling allows cancer cells to escape this signaling, leading to therapeutic resistance. Treatment resistance is a major challenge in the development of effective therapies. The microRNAs (miRNAs) play important roles in CRC treatment resistance and CRC progression and apoptosis. This review discusses the role of miRNAs in contributing to the promotion or inhibition of apoptosis in CRC and the role of miRNAs in modulating treatment resistance in CRC cells.

## 1. Introduction

Colorectal cancer (CRC) is a type of cancer that starts in the colon or rectum. CRC is the third most common cancer diagnosed in the United States. It is also the third leading cause of cancer death in the world, and its incidence is rising in developing countries [1]. Lately, due to progress in screening techniques and improvements in treatment, the death rate from CRC has decreased. Although CRC typically affects older adults, its incidence and mortality are rising among young adults [2].

Environmental and genetic factors play major roles in the pathogenesis of CRC; nutrition plays a causal and protective role in the development of CRC [3]. CRC can also be influenced by other factors such as dietary habits, smoking, a low level of physical exercise, an aging population, and obesity. Among the genetic factors, Sprouty (SPRY) is an intracellular regulator of receptor tyrosine kinase (RTK) signaling. Members of the SPRY family (SPRY1–4) of proteins have been identified as modulators of RTK signaling. SPRY2 functions as a putative oncogene in CRC [4]. The *SMAD7* gene, which is involved in transforming growth factor-beta (TGFβ) signaling, was found to be evolved in CRC [5]. In addition, low socio-economic status was associated with an increased risk of CRC in the US [3,6]. 

The treatment options for CRC include surgery, chemotherapy, radiotherapy, and targeted therapy. Patients with CRC in its earliest stage usually have surgery as the first treatment. Chemotherapy (adjuvant treatment) may also be used after surgery. A combination of two or more treatments is often suggested, depending on the stage of cancer development. Although surgery and chemotherapy have long been the first choices for CRC patients, the prognosis has never been satisfying for metastatic CRC patients. Targeted therapy is a new option that has successfully prolonged the overall survival for CRC patients [7]. Chemotherapy includes both single-agent therapy and multiple-agent regimens. The single-agent therapy used is mainly fluoropyrimidine (5-FU)-based, and multiple-agent regimens include one or several drugs, such as oxaliplatin, irinotecan, and capecitabine. Although 5-FU remains the gold standard of first-line treatment for CRC, relapses frequently occur, indicating the existence of cancer cells that are therapy-resistant [8]. The use of a nano drug delivery system has been discussed to improve the therapeutic outcome of 5-FU [9]. The therapy 5-FU enhances the sensitization of CRC cells to drug-induced apoptosis. The therapy 5-FU might activate caspase-6 to trigger colon cancer cell apoptosis. Resveratrol is a natural polyphenolic compound. The therapy 5-FU and resveratrol combination treatments cause anti-cancer activities by inhibiting STAT3 and Akt signaling pathways, thereby increasing CRC cell apoptosis [10]. 

Apoptosis is the process of programmed cell death that occurs during normal development and aging and is a homeostatic mechanism to maintain cell populations. In normal tissues, there is a balance between the generation of new cells and the loss of cells. Cells frequently die through the apoptosis process. The two main pathways in apoptosis are the extrinsic and intrinsic apoptosis signaling pathways [11,12]. The extrinsic (or death receptor) pathway that initiates apoptosis involves transmembrane-receptor-mediated interactions. These involve death receptors that are members of the tumor necrosis factor (TNF) receptor gene superfamily. In addition, caspases are the primary mediators of apoptosis. The intrinsic pathway of caspase activation is initiated by events such as DNA damage and growth factor withdrawal. These events lead to changes in the integrity of the mitochondrial membrane, which is regulated by BCL-2 family proteins.

Apoptosis is of particular interest to researchers who study cancer. When something goes wrong in a cell, this damaged cell is quickly destroyed by apoptosis, which can prevent the development of cancer. Otherwise, the damaged cell may survive and develop into a cancer cell. Cancer cells can evade apoptosis and continuously divide. The p53 tumor suppressor plays a critical role in protecting normal cells from malignant transformation, and the tumor suppressor gene TP53 is mutated in ~50% of human cancers. The p53 tumor suppressor acts as a major barrier to neoplastic transformation and tumor progression through its unique ability to act as an extremely sensitive collector of stress inputs and to coordinate a complex framework of diverse effector pathways and processes that protect cellular homeostasis and genome stability [13]. Reactivating p53 in cancer cells has been an interesting research area. A specific inhibitor of cap-dependent translation, 4EGI-1, was found to cause an increase in p53 internal ribosomal entry site (IRES) activity to induce cancer cell apoptosis [14]. Mouse models show that genetic reconstitution of the wild type p53 tumor suppression functions rescues tumor growth, indicating that either restoring wt-p53 activity or inhibiting mutant p53 oncogenic activity could be promising cancer therapeutic strategies [15]. The mechanism through which p53 prevents tumor development is known as the induction of apoptotic death in nascent neoplastic cells. However, recently, this concept has been challenged, because gene-targeted mice that lack the critical effectors of p53-induced apoptosis do not develop tumors spontaneously [16]. 

Chemotherapy forces cancer cells to undergo apoptosis by causing DNA damage or cellular distress. However, cancer resistance commonly occurs in chemotherapy and often leads to therapeutic failure. Cancer drug resistance can occur through different mechanisms, including apoptosis suppression [17]. Abnormalities in apoptotic function contribute to both the pathogenesis of CRC and its resistance to chemotherapeutic drugs and radiotherapy [18]. Overexpression of mutated p53 is often connected to resistance to standard medications. MDM2 is a negative regulator of p53. Small molecules that can switch off the activity of MDM2 have been identified, and these MDM2 inhibitors increase the activity of combination treatment with standard chemotherapy [19]. The perspective that drug ineffectiveness results from tumor–host interactions was proposed, and it is understood that such an interaction might open new opportunities to overcome the development of resistance to cancer chemotherapy [20].

## 2. microRNA

Epigenetic mechanisms, including DNA methylation and microRNA (miRNA) regulation, play important roles in cancer apoptosis and cancer drug resistance. DNA methylation in cancer has generally been associated with drug resistance and treatment prognosis [21]. It also involves apoptotic signaling and affects the chemotherapy response in CRC [22]. Methylation of the Bim gene, a BH3 only proapoptotic member of the BCL-2 family, is associated with CRC development, metastasis, and chemosensitivity [23]. Furthermore, miRNAs are also important epigenetic factors involved in cancer apoptosis and cancer drug resistance. In this review, we focus on miRNA functions in promoting or inhibiting apoptosis in CRC and their roles in drug resistance.

The miRNAs are small, non-coding RNAs of about 21–24 nucleotides in length, with important functions in cell differentiation, development, cell cycle regulation, and apoptosis [24]. The dysregulation of miRNAs has either a tumor suppressor or oncogene function [25]. It plays a role in tumorigenesis by regulating some oncogenes and tumor suppressor genes. Tumor suppressor miRNAs can down-regulate different oncogenes to promote cell proliferation, invasion, and metastasis. The miRNAs are also regulated by oncogenes and tumor suppressor genes [26]. The first miRNA was discovered in the early 1990s while studying development in the nematode *Caenorhabditis elegans* regarding the gene lin-14 [27]. The miRNAs are involved in mRNA degradation by binding to 3′-untranslated regions (3′UTR), and a single miRNA can bind and regulate the expression of more than 100 different transcripts. Thus, it has been estimated that miRNAs may regulate up to 30% of the protein-coding genes in the human genome [24,28].

The biogenesis of miRNA is classified into canonical and non-canonical pathways. The canonical biogenesis pathway is the dominant pathway. In this pathway, a primary miRNA transcript (pri-miRNA) is cleaved by the endoRNase Drosha in the nucleus to excise the precursor miRNA (pre-miRNA), which is exported to the cytoplasm. The pre-miRNA has a characteristic hairpin secondary structure that is recognized and cleaved in the cytoplasm by the endoRNase Dicer, processing it into the mature miRNA [29]. The miRNAs are synthesized from primary miRNAs in two stages by the action of two RNase III-type proteins: Drosha in the nucleus and Dicer in the cytoplasm [30,31]. The RNase III enzyme Drosha interacts with DGCR8 to initiate miRNA maturation in the nucleus. There are multiple non-canonical miRNA biogenesis pathways that have been elucidated. These pathways use different combinations of the proteins involved in the canonical pathway, and generally, the non-canonical pathways can be grouped into Drosha/DGCR8-independent and Dicer-independent pathways [30]. However, a study revealed that (i) a Drosha truncation that was only located in the cytoplasm could cleave pri-miRNA-like reporters in a DGCR8-dependent manner; (ii) in vitro transcribed pri-miRNA transcripts were processed into mature miRNAs without entering the nucleus; and (iii) a subset of pri-miRNAs in the cytoplasm [32]. This supports the existence of cytoplasmic Drosha activity. In addition, although miRNAs have well-characterized roles in cytoplasmic gene regulation, they have also been implicated in transcriptional gene regulation restricted to the cell nucleus [33]. The miRNAs are differentially expressed in the nucleus and cytoplasm in response to hypoxic stress, which offers new insight into the molecular response to hypoxia. Widespread differential miRNA expression in the nucleus suggests that miRNAs are likely to perform extensive regulatory functions in the nucleus [33].

The miRNA nucleotide sequences can be accessed at the databases miRBase or MirGeneDB [34,35,36,37]. The miRBase is the primary online resource for microRNA sequences. It provides a wide range of information on published miRNAs, including sequences, biogenesis precursors, genome coordinates and context, literature references, deep sequencing expression data, and community-driven annotations. The latest release of miRBase contains miRNA sequences from 271 organisms, including 38,589 hairpin precursors and 48,860 mature microRNAs [34]. In contrast to miRBase, another database, MirGeneDB, focuses on the identification of miRNA genes and families, rather than sequences. MirGeneDB contains annotations of 10,899 bona fide and consistently named miRNAs, constituting 1275 miRNA families from 45 species that represent every major metazoan group [37].

The miRNAs are especially involved in the initiation and progression of cancers [38,39]. Calin et al. published the first study to link two miRNAs, miR-15 and miR-16, to cancer in 2002. Their study focused on B-cell chronic lymphocytic leukemia (B-CLL) [40]. After that, many studies showed that miRNAs can initiate carcinogenesis or drive the progression of cancer [41,42,43,44]. The miR-527 acts as an oncogenic miRNA in esophageal squamous cell carcinoma by directly targeting PHLPP2 [42]. LINC02381 acts as an oncogene in the development of cervical cancer that promotes cell viability and metastasis via miR-133b [43]. The miR-1193 plays a tumor inhibitory role in cervical cancer by directly targeting CLDN7 [44]. The miR-3174 inhibits the proliferation of bladder cancer cells by targeting ADAM15 [45]. Obesity is associated with insulin resistance, which is a risk factor for CRC development. The miRNAs have been shown to be involved with insulin sensitivity, glucose tolerance, and lipid metabolism, in both obesity and CRC [46]. There is a close functional connection between miRNAs and host genes. Exploration of this relationship had added to the understanding of the transcriptional and post-transcriptional regulation of miRNAs and the development of miRNA therapeutics in cancer [47]. In addition to cancer, miRNAs may also contribute to neurological diseases and inflammation. The miR-34a and the miR-504, as well as other miRNAs, have been identified to be associated with amyotrophic lateral sclerosis (ALS) [48,49]. The miRNAs are also related to the development of Parkinson’s disease (PD) [49,50], frontotemporal dementia [51], Alzheimer’s disease [52], and spinal muscular atrophy [53]. Additionally, let-7b is a miRNA biomarker for anti-NMDA receptor encephalitis [54,55,56]. In addition to being a useful disease biomarker, miRNA can be used to explore the association between two diseases or between disease and vaccination. Many diseases may have common miRNA biomarkers. Exploring the relationship of their miRNA biomarkers may provide a useful approach for investigating their association [57,58,59].

Drug resistance is one of the main obstacles in cancer chemotherapy. Over 90% of the mortality of patients with cancer is connected to drug resistance [60]. In recent years, miRNAs have been shown to be involved in the cancer drug resistance through the evasion of apoptosis, cell cycle alterations, and drug target modifications. The connection between apoptosis and drug resistance is complex, and not all drug resistance mechanisms can be attributed to apoptosis derangement. A retrospective study was conducted to measure the miRNA expression profiles of patients with adenocarcinomas of the esophagogastric junction who received neoadjuvant chemotherapy [61]. Several miRNAs were identified as predictors of chemoresistance or contributors to chemosensitivity. Resistance to Cisplatin-Fluorouracil (CF) combination therapy is the main obstacle in the treatment of gastric cancer. The contribution of miRNAs to the development of resistance against CF chemotherapy has been demonstrated [60,62]. The miRNAs mainly affect head and neck cancer (HNC) drug resistance by influencing the expression of target genes and by regulating apoptosis pathways, epithelial–mesenchymal transformation, and cancer stem cells [63]. The miRNAs play a role in chemotherapeutic resistance in leukemia, and the deregulation of miRNAs is associated with resistance to chemotherapy in leukemia [64]. The miRNAs regulate key genes responsible for drug resistance in ovarian cancer. The miR-363 regulates the ATP-binding cassette subfamily B member 1 gene (ABCB1) in the paclitaxel-resistant cell line, and miR-29a regulates the collagen type III alpha 1 chain gene (COL3A1) in the topotecan-resistant cell line [65].

## 3. miRNA in Apoptosis of Colorectal Cancer

Many miRNAs are involved in the apoptosis pathway in CRC. Table 1 lists the miRNAs that contribute to the promotion or inhibition of apoptosis in CRC cells and lists the miRNAs that are correlated with drug resistance in colorectal tumors. These miRNAs could be useful biomarkers for the development of diagnosis, prognosis, therapy prediction, and therapeutic tools for CRC.

A study performed a copy number analysis of the six mature miRNAs in the miR-17–92 cluster in colon tumor tissues: miR-17, miR-18a, miR-19a, miR-20a, miR-19b, and miR-92a [66]. The result showed that miR-92a was transcribed at higher levels than the other five miRNAs in both carcinoma and adenomas. Additionally, miR-92a targeted the anti-apoptotic molecule BCL-2-interacting mediator of cell death in colon cancer tissues. Abrogation of miR-92a induces apoptosis in cancer cells, that is, the decrease in miR-92a leads to apoptosis in cancer cells. It was observed that cells transfected with the anti-miR-92a antagomir, but not with miR-92a, underwent cell death. As a result, it was determined that miR-92a plays a pivotal role in the development of colorectal carcinoma. A study compared dysregulated apoptosis genes with differentially expressed miRNAs to identify the influence of miRNAs on apoptosis in CRC [67]. The miR-92a is associated with the two apoptosis-related genes, CSF2RB and BCL2L1, in the Kyoto Encyclopedia of Genes and Genomes (KEGG) apoptosis pathway [67]. Increased expression of miR-92a-3p in carcinoma tissue can improve the survival of patients. CSF2RB was shown to be related to miR-92a with a negative beta coefficient, suggesting a greater likelihood for direct binding that would alter the gene expression.

Overexpression of miR-766 was found to reduce cell growth and induce apoptosis in colon cancer cells through suppression of the MDM4/p53 pathway. On the contrary, the downregulation of miR-766 was shown to promote cell growth and reduce apoptosis in colon cancer cells [68]. Ectopic expression of miR-100 inhibits cell growth and induces apoptosis, whereas knockdown of miR-100 leads to the reverse phenotype [69]. Overexpression of miR-378-5p was found to induce cell cycle arrest and promote apoptosis in CRC cells. In addition, miR-378-5p was shown to down-regulate gene BRAF in CRC cells [54]. Ectopic expression of miR-18a was found to inhibit the repair of DNA damage induced by etoposide, leading to the accumulation of DNA damage and increasing colon cancer cell apoptosis [70]. The transfection of miR-18a was shown to induce apoptosis in colon cancer cell lines (SW620 cells) by directly binding to oncogenic heterogeneous nuclear ribonucleoprotein A1 (hnRNP A1) [71]. Overexpression of miR-125a-5p was shown to inhibit cell proliferation and induce apoptosis in colon cancer cells. The anti-apoptotic genes BCL2, BCL2L12, and Mcl-1 are direct targets of miR-125a-5p, and they were shown to be down-regulated by miR-125a-5p overexpression [72]. miR-125b may promote apoptosis in CRC cell lines by suppressing the anti-apoptotic molecules of the BCL-2 family, and miR-125b down-regulation may facilitate tumor development [73]. Silencing miR-200c expression led to the upregulation of phosphatase and the tensin homolog (PTEN) protein and p53 Ser15 phosphorylation levels in color cancer cell lines. The miR-200c functions as an oncogene in colon cancer cells by regulating tumor cell apoptosis, survival, invasion, and metastasis through the inhibition of PTEN expression and p53 phosphorylation [74].

Baicalin is a component extracted from the roots of *Scutellaria baicalensis*. Baicalin treatment has been demonstrated to inhibit the proliferation of colon cell lines. By inhibiting DKK1, miR-217 was shown to promote apoptosis and reduce the survival rate of colon cells induced by Baicalin [75]. Ectopic miR-206 in colon cancer cell lines (LOVO and SW620) was found to increase the rate of apoptosis, while depletion of miR-206 in HCT116 cells had the reverse effect, revealing that miR-206 may inhibit cell proliferation by arresting tumor cells at the G1/G0 phase and accelerating apoptosis [76]. When the colon cancer cell lines HCT116 and Caco-2 were treated with PGE2, the percentage of apoptotic cells decreased. This demonstrates that PGE2 regulates CRC cell proliferation and apoptosis, and miR-206 may induce apoptosis by targeting TM4SF1 in PGE2-induced cells [77]. Upregulation of miR-206, targeting NOTCH3, inhibited cancer cell proliferation and activated apoptosis in colon cell lines [78]. The miR-210 was found to be associated with the upregulation of pro-apoptotic Bim expression, thereby mediating the induction of apoptosis. Overexpression of miR-210 induced reactive oxygen species (ROS) generation and apoptosis in CRC, which was associated with the upregulation of pro-apoptotic Bim expression and Caspase 2 processing [79].

The miR-129 regulates BCL2 expression, leading to the activation of the intrinsic apoptosis pathway [80]. By targeting BMI1 polycomb ring finger oncogene, miR-218 was found to inhibit cell cycle progression and promote apoptosis in colon cancer [81]. The miR-218 and caspase-8 expressions were found to decrease while c-FLIP expression was elevated in human colon cancer tissues [82]. The miR-195 was shown to promote apoptosis in colorectal cancer cell lines by targeting antiapoptotic Bcl-2 [83]. Bcl-X(L) is a direct target of miR-491, and its silencing was found to contribute to miR-491-induced apoptosis in human colorectal cancer cells [84]. The miR-7 was shown to be downregulated in CRC cell lines. It targets the 3′ untranslated region of XRCC2, inducing apoptosis in CRC [85]. The miR-148a was found to inhibit Bcl-2, leading to the activation of an intrinsic mitochondrial pathway and tumor apoptosis in CRC [86].

The miR-708 was shown to suppress cell proliferation, induce apoptosis, and reduce metastasis in CRC in vitro by directly targeting ZEB1 through the AKT/mTOR signaling pathway [87]. Annexin/PI staining and caspase-3/PARP activation demonstrated that miR-182 treatment significantly increases apoptosis in the CRC cell line [88]. SIRT1 may be a target of miR-34a that contributes to its ability to promote apoptosis in the CRC cell line [89]. Ectopic expression of miR-133b was found to inhibit cell proliferation caused cell cycle arrest in the G1 phase, and it induced apoptosis in CRC cells by direct targeting of the MET receptor tyrosine kinase [90]. Silencing of DFF45 by miR-145 accounts, at least in part, for staurosporine-induced colon cancer cell apoptosis in vitro [90]. The miR-143 was shown to be reduced in colon cancer and the introduction of miRNA-143 into colon cancer cells induced apoptosis and slower growth in a xenograft model [91]. The miR-143 was shown to significantly reduce human colon cancer cell xenograft growth in vivo, leading to increased tumor cell apoptosis and decreased proliferation [92].

In a previous study, miR-342 reconstitution resulted in a marked increase in apoptosis, whereas transfection of negative control miRNA mimics did not affect CRC cells [93]. Overexpression of miR-26b led to significant suppression of cell growth, induction of apoptosis in the CRC cell line in vitro, and inhibition of tumor growth in vivo [92]. Overexpression of miR-630 in CRC cell lines (HT29 and SW480) increased cancer cell apoptosis and death in vitro by targeting BCL2L2 and TP53RK [94]. APC/Wnt/β-catenin signaling has been linked to reduced apoptosis, and Adenomatous Polyposis Coli (APC) re-expression was found to cause apoptosis in colon cancer cell lines [95,96]. Re-expression of APC was shown to cause apoptosis by downregulating miR-135b [97]. The miR-532-3p was found to downregulate in CRC and function as a sensitizer for chemotherapy in CRC by inducing apoptosis through activating effects on p53 and apoptotic signaling pathways [98]. The miR-769 was shown to downregulate in CRC and target CDK1 in the regulation of CRC cell proliferation, apoptosis, migration, and invasion. Recovery of miR-769 expression was found to increase CRC cell apoptosis in vitro and inhibit tumor growth in vivo [99]. The tumor necrosis factor-related apoptosis-inducing ligand (TRAIL) was associated with the expression level of miR-20a in CRC. miR-20a regulated BID, which is a pro-apoptotic member of the BCL-2 family, in CRC cells. The knockdown of miR-20a was shown to induce the mitochondrial pathway of apoptosis by inhibiting the translocation of tBID to the mitochondria [100]. Exogenous miR-21 over-expression mimicked the effect of RhoB knockdown in promoting proliferation and invasion and inhibiting apoptosis in CRC cells [101]. The miR-21 loss reduced STAT3 and BCL-2 activation, causing an increase in the apoptosis of tumor cells in colitis-associated colon cancer mice [102]. The miR-32 inhibited apoptosis by directly targeting OTUD3 in colon cancer cells. The miR-32 upregulation reduced cell apoptosis, while its downregulation displayed opposite effects [103]. The decrease in FAS expression could contribute to the reduction of apoptosis in CRC cells. Anti-miR-196b was shown to up-regulate FAS expression and increased apoptosis in CRC cell lines [104].

5-FU is a classic chemotherapeutic drug that has been widely used for CRC treatment. The overexpression of miR-96 decreases the expression of the anti-apoptotic regulator XIAP and the p53 stability regulator UBE2N, resulting in increased apoptosis and growth inhibition following 5-FU exposure. As a result, miR-96 may modulate 5-FU sensitivity in CRC cells by promoting apoptosis [105]. Restoration of miR-365 expression was shown to inhibit cell cycle progression and promote 5-FU-induced apoptosis in colon cancer cell lines. The antitumor effects of miR-365 are likely mediated by targeting Cyclin D1 and Bcl-2, thus inhibiting cell cycle progression and promoting apoptosis [106]. In 5-FU-treated colon cancer cells, the expression of miR-23a was found to increase and the level of apoptosis-activating factor-1 (APAF-1) decreased compared with untreated cells. The miR-23a antisense was shown to enhance 5-FU-induced apoptosis through the APAF-1/caspase-9 apoptotic pathway [107]. Downregulation of miR-23a promoted cell apoptosis in microsatellite instability (MSI) CRC cells treated with 5-FU. The miR-23a was found to enhance 5-FU resistance in MSI CRC cells by targeting ABCF1 [108]. The forced expression of miR-21 significantly inhibited apoptosis, enhanced proliferation and invasion, and increased the resistance of tumor cells to 5-FU and X radiation in HT-29 colon cancer cells [109]. The overexpression of miR-21 was correlated with 5-FU drug resistance in colorectal tumors, and miR-21 was found to downregulate the core mismatch repair proteins hMSH2 and hMSH6, leading to a defect in damage-induced G2/M arrest and apoptosis. As a result, miR-21 is likely to be a useful marker for therapeutic protocols in CRC [110]. In an in vitro experiment using two CRC cell lines, LoVo and SW480, miR-17-5p expression levels decreased in cells treated with 5-Fu. In addition, apoptosis assays confirmed that miR-17-5p suppressed apoptosis [111]. BIM is a member of the BCL-2 homology 3(BH3)-only subgroup of the BCL-2 family that potently induces apoptosis [112]. In vitro studies showed that miR-10b directly inhibits BIM, and the overexpression of miR-10b led to chemoresistance in CRC cells to 5-FU [113]. Ectopic expression of miR-520g conferred resistance to 5-FU-induced apoptosis by inhibiting p21 expression [114]. The miR-22 was shown to inhibit autophagy and promote apoptosis to regulate the 5-FU treatment sensitivity of CRC cells through post-transcriptional silencing of BTG1 [115].

## 4. Discussion

As such, miRNA regulation is one of the fundamental processes driving CRC initiation and progression. Many miRNAs have been identified as promotors of apoptosis or inhibitors of apoptosis in CRC. In addition to the inhibition of cancer cell proliferation and the activation of apoptosis, miRNA also functions as a tumor suppressor in CRC. In the first study of miRNAs in colon cancer, Connor et al. identified miR-143 and miR-145 as novel dysregulated miRNAs in colon cancer [121]. Furthermore, miR-155 and miR-21 expression levels are significantly correlated with CRC [122], and miR-1 functions as a tumor suppressor in CRC by downregulating the MET oncogene [123]. The miR-92a has been reported to be an oncogene in CRC. As a result, miRNA has become a useful biomarker for early detection of CRC in both serum and stools [124].

The miRNAs can interact with signaling pathways to contribute to the development of CRC. The APC/beta-catenin pathway is known to play a crucial role in sporadic colorectal carcinogenesis [125]. The Wnt-signaling pathway functions in regulating cell growth in CRC by inactivating mutations in the APC gene [126,127]. Most sporadic and familial CRCs involve hyperactivation of the Wnt-signaling pathway. Mutational inactivation of the APC tumor suppressor might be the initiating event in most CRCs. Wnt hyperactivation is the key oncogenic driver in most CRCs, and disruption of APC drives the activation of the Wnt-signaling pathway [128]. An important CRC drug target is the phosphatidyl-inositol 3-kinase (PI3K) pathway, which is frequently deregulated in patients with CRC [129]. A large proportion of human colon tumors possess mutations in the APC gene or β-catenin gene which affect downstream signaling of the PI3K/Akt pathway [130]. Chronic inflammation is a risk factor for CRC. The cytokines IL-6 and TNF-α and transcription factors, STAT3 and NFKB contribute to both inflammation and cancer development [10]. The miRNAs modulate inflammatory pathways mediated by NFΚB and STAT3 in CRC [127]. The transforming growth factor-beta (TGF-β) signaling pathway plays pivotal roles in cell development, proliferation, differentiation, and apoptosis, thereby promoting CRC formation, invasion, and metastasis [131].

Circulating miRNAs are emerging as promising biomarkers for the early screening, prognosis, and treatment of CRC. A cohort of 163 samples representing healthy control participants, patients with benign adenomas, and patients with CRC in independent discovery and validation sets was studied to find potential miRNA biomarkers [132]. Three miRNA ratios (miR-17-5p/miR-135b, miR-92a-3p/miR-135b, and miR-451a/miR-491-5p) were identified to be significantly upregulated in adenoma patients compared with the healthy control group. Five miRNA ratios (let-7b/miR-367-3p, miR-130a-3p/miR-409-3p, miR-148-3p/miR-27b, miR-148a-3p/miR-409-3p, and miR-21-5p/miR-367-3p) were effective at distinguishing patients with CRC from the benign adenoma and healthy control groups.Circulating small extracellular vesicles (sEVs) have a distinct miRNA profile (let-7b-3p, miR-139-3p, miR-150-3p, miR-145-3p) in colon cancer patients at an early stage compared with non-cancerous controls; therefore, they are proposed as a new promising biomarker category for early CC patients [133]. Significant upregulation of miR-20a was demonstrated in the serum of CRC patients, tumor tissues, and cell lines [100]. This reveals that circulating miRNA could be a potential CRC diagnosis biomarker.

## 5. Conclusions

CRC is the third most deadly cancer in the world, and its incidence has been steadily rising worldwide. The use of miRNAs to modify the signaling pathways involved in the initiation, progression, and apoptosis of CRC has been studied, as well as the role of miRNAs as oncogenes or tumor suppressors. In addition, their role in increasing apoptosis and growth inhibition following 5-FU exposure or in conferring resistance to 5-FU-induced apoptosis has been studied. Treatment with 5-FU is known to improve survival in various cancers, especially CRC. The therapy 5-FU is the main first-line treatment for CRC, but its antineoplastic activity is limited in drug-resistant cancer cells. In this paper, miRNAs were shown to contribute to the promotion or inhibition of apoptosis in CRC, and we reviewed the correlation of miRNAs with 5-FU drug resistance. The miRNAs can act as potential biomarkers for molecular therapy in CRC. Since many miRNAs are involved in the CRC mechanism, the integration of multiple miRNA biomarkers to improve the diagnosis and treatment of CRC warrants further investigation.

## Figures and Tables

**Table 1 ijms-21-05353-t001:** miRNA mediates the apoptosis pathway.

Colon Cancer, Colorectal Carcinoma		
miRNA	Gene or Protein	Reference
miR-92a	BCL-2, CSF2RB, BCL2L1	[66,67]
miR-766	MDM4 p53/Bax signaling pathway	[68]
miR-21	hMSH2, hMSH6, PDCD4, TIAM1, SPRY2, PTEN, TGFBR2, CDC25A, hMSH2, RhoB, STAT3, Bcl-2, BIRC5	[67,101,102,109,110,116,117]
miR-96	TP53INP1, FOXO1, FOXO3A, UBE2N, XIAP, REV1, RAD51	[105,116]
miR-17	P130, BIRC5	[67,111]
miR-100	RAP1B	[69]
miR-365	Cyclin D1, Bcl-2	[106,117]
miR-378	BRAF	[54]
miR-18a	ATM, hnRNP A1	[70,71]
miR-125a	BCL-2, BCL2L12 and Mcl-1	[72]
miR-125b	IL-6R	[73]
miR-10b	BIM (BCL2L11)	[113]
miR-200c	PTEN expression and p53 phosphorylation	[74]
miR-217	DKK1	[118]
miR-206	NOTCH3, FMNL2, TM4SF1	[76,77,78]
miR-210	Bim, Mcl-1	[79]
miR-23a	ABCF1, APAF-1	[107,108]
miR-520g	p21	[114]
miR-129	BCL2	[80]
miR-32	OTUD3	[103]
miR-218	BMI1, c-FLIP	[81,82]
miR-195	BCL2, BIRC5,TUBA1B	[67,83]
miR-491	Bcl-X(L)	[84]
miR-7	XRCC2	[85]
miR-148a	BCL2	[86]
miR-708	ZEB1(Akt/mTOR signaling pathway)	[87]
miR-182		[88]
miR-34a	SIRT1, Par-4, p53	[89,119]
miR-133b	c-Met	[90]
miR-145	DFF45, BIRC5	[67,90]
miR-143	Evi 1 (PI3K/Akt pathway)	[91,92]
miR-342		[93,120]
miR-26b		[92]
miR-630	TP53RK, BCL2L2	[94]
miR-135b	APC	[97]
miR-196b	FAS, TNFRSF10B	[67,104]
miR-22	BTG1	[115]
miR-532	P53	[118]
miR-769	CDK1	[99]
miR-20a	BCL2	[100]

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
