# Peer review of "MicroRNAs and Apoptosis in Colorectal Cancer"

_ijms, 2020, doi:10.3390/ijms21155353_

Round 1
Reviewer 1 Report
The authors presented a review entitled "MicroRNAs and apoptosis in colorectal cancer" and present the role of miRNAs in contributing to promote or inhibit apoptosis in CRC and their role on modulating treatment resistance in CRC cells.
The work is very interesting and can be a useful tool for its readers in order to expand their knowledge of the subject matter. However, in some parts it is not very clear and not easy to understand and a more accurate revision of English is also recommended.
In my opinion the work can be accepted with minor revision.
to follow some of my small observations :
14) line 243: you wrote "miR133b" instead of "miR-133b" 15) Chapter: "miRNA in apoptosis of colorectal cancer", from line 166 to 286: the chapter is well deepened, but it appears slightly confused and not very structured, I recommend following a more structured scheme. Sometimes you start speaking about a miRNA without an introduction. 16) line 293: you wrote for the first time about miR-1, why you do not speak about this in the chapter "miRNA in apoptosis of colorectal cancer", and do not indicate it in table 1? 17) line 313: you wrote "miR135b" instead of "miR-135b"
Author Response
Thank you very much for the valuable comments. I have prepared a revised version incorporating with the comments. The point-by-point responses are as follows. This version has been edited by MDPI English service. The certificate of English editing is attached.
- line 15 and line 95 : Please change "contributing to promote apoptosis or inhibit apoptosis" in " contributing to promote or inhibit apoptosis"
Response: This version was edited by MDPI English service. These two places are changed as follows.
Line 15. It was changed to “contributing to the promotion or inhibition of apoptosis”
Line 97. It was changed to “promoting or inhibiting apoptosis".
2) from line 25 to line 32: The authors speak before about nutrition, than about SPRY and SMAD7, and after restart speaking about nutrition. In mine opinion is better to speak first about an argument and then start with the other argument. Furthermore it is better to introduce the genetic factors, for example "Among the genetic factors, Sprouty (SPRY)..."
Response: Lines 27 and 28. I moved the sentence “CRC can also be influenced by other factors such as dietary habits, smoking, a low level of physical exercise, an aging population, and obesity” to the place just after the first sentence about nutrition. Then all the sentences about nutrition are placed together.
Line 29. I added “Among the genetic factors” before “Sprouty (SPRY)….”.
3) line 34, 35: Please could remodel the sentence making it smoother and clearer: "Patients with CRC that have not spread to distant sites usually have surgery as the first treatment" ?
Response: Line 36. This sentence was revised to “Patients with CRC in its earliest stage usually have surgery as the first treatment.”.
4) line 68: Please introduce 4EGI-1; could you please write the full name of IRES?
Response: Line 70. “4EGI-1” was changed to “A specific inhibitor of cap-dependent translation, 4EGI-1,”.
Line 71. The full name of IRES “internal ribosomal entry site” was added.
5) line 76, 77: Please could remodel the sentence making it smoother and clearer: "The mechanism that p53 prevents tumor development is known as the induction of apoptotic death in nascent neoplastic cells." ?
Response: Line 74. This sentence was revised to “The mechanism through which p53 prevents tumor development is known as the induction of apoptotic death in nascent neoplastic cells.”.
6) line 93: probably it is better to speak about microRNAs in plural, as a class of molecules
Response: Line 95. This sentence was revised to “miRNAs are also important epigenetic factors”.
7) line 96-99: could you, it would be interesting, for the less experienced, to introduce some bibliographic references of this part
Response: Lines 99 and 100. I added two references [24-25] for this part.
11) line 167, 168: please check the english of: "Table 1 lists the miRNAs contributing to promote apoptosis or inhibit apoptosis in CRC cells".
Response: Lines 181-183. This sentence was revised to “Table 1 lists the miRNAs that contribute to the promotion or inhibition of apoptosis in CRC cells and lists the miRNAs that are correlated with drug resistance in colorectal tumors.”
12) line 172: you wrote "miR17-92" instead of "miR-17-92"
Response: Line 185. It was revised to “miR-17-92".
13) line 222: after "apoptosis", please insert puntaction.
Response: Line 229. A period was inserted after "apoptosis".
14) line 243: you wrote "miR133b" instead of "miR-133b"
Response: Line 246. It was revised to “miR-133b ".
15) Chapter: "miRNA in apoptosis of colorectal cancer", from line 166 to 286: the chapter is well deepened, but it appears slightly confused and not very structured, I recommend following a more structured scheme. Sometimes you start speaking about a miRNA without an introduction.
Response: Lines 180-304. I have revised this section by moving the part about 5-FU to the last two paragraphs such that the section can be more structured.
16) line 293: you wrote for the first time about miR-1, why you do not speak about this in the chapter "miRNA in apoptosis of colorectal cancer", and do not indicate it in table 1?
Response: miR-1 functions as a tumor suppressor in CRC, but may not involve in apoptosis pathway of CRC. Therefore, I did not write it in the section "miRNA in apoptosis of colorectal cancer" and did not include in Table 1.
17) line 313: you wrote "miR135b" instead of "miR-135b"
Response: It was revised to “miR-135b "
Reviewer 2 Report
This is a useful contribution to the field as a review of the role of miRNAs in colorectal cancer, with an emphasis on apoptosis. The strengths of the paper are that is relatively well written, covers the indicated topic well, and can be used a resource for researchers in the field, particularly with respect to utilizing miRNAs for biomarkers for drug resistance, treatment efficacy, progression,etc or even identifying relevant miRNAs for targeted for therapeutics. Weaknesses, which should be the focus of a minor revision include:
- In the background material on miRNAs, we read: "miRNA are involved in mRNA degradation by binding to 3’-untranslated regions (3’UTR), and a single miRNA can bind and regulate the expression of more than 100 different transcripts." Later on,the author talks about some newly discovered nuclear functions of miRNAs. Now, that's all true and fine as it goes, but miRNAs function to regulate gene expression by a variety of mechanisms, and likely foremost in mammalian cells is via repression of translation. A review about miRNAs needs to at least mention the major mechanisms of miRNA function and emphasize those that are predominant in human cells.
2. The Wnt pathway is not only activated by APC mutation (loss of function), but in a minority of Wnt-positive tumors by gain of function oncogenic beta-catenin mutation.
3. The author does briefly mention miRNAs associated with APC/Wnt, but given the importance of this pathway in CRC, is there anything else?
4. Given the importance of nutrition/diet with respect to CRC, could there be more insights - if any are known - how diet and other lifestyle issues (obesity?) affect miRNAs that influence CTC?
Author Response
Thank you very much for the valuable comments. I have prepared a revised version incorporating with the comments. The point-by-point responses are as follows. This version has been edited by MDPI English service. The certificate of English editing is attached.
1.In the background material on miRNAs, we read: "miRNA are involved in mRNA degradation by binding to 3’-untranslated regions (3’UTR), and a single miRNA can bind and regulate the expression of more than 100 different transcripts." Later on,the author talks about some newly discovered nuclear functions of miRNAs. Now, that's all true and fine as it goes, but miRNAs function to regulate gene expression by a variety of mechanisms, and likely foremost in mammalian cells is via repression of translation. A review about miRNAs needs to at least mention the major mechanisms of miRNA function and emphasize those that are predominant in human cells.
Response: I included the biogenesis of miRNA.
Lines 109-114. The biogenesis of miRNA is classified into canonical and non-canonical pathways. The canonical biogenesis pathway is the dominant pathway. In this pathway, a primary miRNA transcript (pri-miRNA) is cleaved by the endoRNase Drosha in the nucleus to excise the precursor miRNA (pre-miRNA), which is exported to the cytoplasm. The pre-miRNA has a characteristic hairpin secondary structure that is recognized and cleaved in the cytoplasm by the endoRNase Dicer, processing it into the mature miRNA [29]
Lines 116-120. There are multiple non-canonical miRNA biogenesis pathways that have been elucidated. These pathways use different combinations of the proteins involved in the canonical pathway, and generally, the non-canonical pathways can be grouped into Drosha/DGCR8-independent and Dicer-independent pathways [30].
- The Wnt pathway is not only activated by APC mutation (loss of function), but in a minority of Wnt-positive tumors by gain of function oncogenic beta-catenin mutation.
Response: Lines 259-260 , we have mentionedβ-catenin in this sentence “APC/Wnt/β-catenin signaling has been linked to reduced apoptosis, and Adenomatous Polyposis Coli (APC) re-expression was found to cause apoptosis in colon cancer cell lines [96, 97]”. In addition, I include more concepts of Wnt- signaling pathway in the Discussion section (see response for Comment 3).
- The author does briefly mention miRNAs associated with APC/Wnt, but given the importance of this pathway in CRC, is there anything else?
Response: I include more concepts about Wnt- signaling pathway.
Lines 319-322. Most sporadic and familial CRCs involve hyperactivation of the Wnt-signaling pathway. Mutational inactivation of the APC tumor suppressor might be the initiating event in most CRCs. Wnt hyperactivation is the key oncogenic driver in most CRCs, and disruption of APC drives activation of the Wnt-signaling pathway [129].
- Schatoff, E.M., B.I. Leach, and L.E. Dow, Wnt Signaling and Colorectal Cancer. Curr Colorectal Cancer Rep, 2017. 13(2): p. 101-110.
- Given the importance of nutrition/diet with respect to CRC, could there be more insights - if any are known - how diet and other lifestyle issues (obesity?) affect miRNAs that influence CTC?
Response: We include these sentences and a reference for this issue.
Lines 147-149. Obesity is associated with insulin resistance, which is a risk factor for CRC development. miRNAs have been shown to be involved with insulin sensitivity, glucose tolerance, and lipid metabolism, in both obesity and CRC [46].
- Cirillo, F., et al., Obesity, Insulin Resistance, and Colorectal Cancer: Could miRNA Dysregulation Play a Role? International Journal of Molecular Sciences, 2019. 20(12).
Reviewer 3 Report
This is a review on the role of miRNA in colorectal cancer with the special emphasisi on apoptosis. While the sugbect is certainly worth reviewing, the manuscript is not well organised and has substantial problems:
1. there are some statements which are untrue or unsupported such as
- The treatment options for CRC include surgery, chemotherapy, and radiotherapy ... they also include many targeted treatment
- Over 90% of the mortality of patients with cancer is connected to drug resistance..what about surgical or radiotherapy failures?
- The contribution of miRNAs in the development of resistance
against CF chemotherapy has been demonstrated..they cite a review on gastric cancer, not the original report
2. Lines 130 and onwards contain many information that are not relevant to the topic (other cancers, neurological disease...)
3. The connection between apoptosis and drug resistance is complex, and not all drug resitance mechanisms can be equated with apoptosis derangement
4. Table 1 should provide brief information on the role of individual listed miRNAS
5. Discussion could be renamed as Conclusion, and the Conclusion paragraph deleted as it is not related to the topic
6. Language editing is needed.
Author Response
Thank you very much for the valuable comments. I have prepared a revised version incorporating with the comments. The point-by-point responses are as follows. This version has been edited by MDPI English service. The certificate of English editing is attached.
- there are some statements which are untrue or unsupported such as
The treatment options for CRC include surgery, chemotherapy, and radiotherapy ... they also include many targeted treatment
Over 90% of the mortality of patients with cancer is connected to drug resistance..what about surgical or radiotherapy failures?
The contribution of miRNAs in the development of resistance
against CF chemotherapy has been demonstrated..they cite a review on gastric cancer, not the original report
Response: The sentence “The treatment options for CRC include surgery, chemotherapy, and radiotherapy.” was revised to “The treatment options for CRC include surgery, chemotherapy, and radiotherapy and targeted therapy.”. In fact, in the original version, I have mentioned targeted therapy in this sentence “Targeted therapy is a new option that has successfully prolonged overall survival for CRC patients [7].”. But I forgot to mention in the sentence about the treatment options in the original version. Thank you for pointing out this.
About the CRC surgical or radiotherapy failures, I have checked literature including the following two references and others. They did not specify the mortality of CRC patients due to surgical or radiotherapy failures. Therefore, I did not include it in this revision.
ZHANG, Yumei; CHEN, Zhiyu; LI, Jin. The current status of treatment for colorectal cancer in China: A systematic review. Medicine, 2017, 96.40.
GENG, Liying; WANG, Jing. Molecular effectors of radiation resistance in colorectal cancer. Precision Radiation Oncology, 2017, 1.1: 27-33.
The original report was cited as reference [63].
Si, W.G., et al., The role and mechanisms of action of microRNAs in cancer drug resistance. Clinical Epigenetics, 2019. 11.
- Lines 130 and onwards contain many information that are not relevant to the topic (other cancers, neurological disease...)
Response: This section introduces miRNA biogenesis, new development and miRNA biomarkers. Since miRNAs are well known as biomarkers for many diseases not limited to CRC, I include some general information for miRNA biomarkers in this section including other diseases.
- The connection between apoptosis and drug resistance is complex, and not all drug resitance mechanisms can be equated with apoptosis derangement
Response: I agree with it and have added this sentence in the text.
Lines 164 and 165. The connection between apoptosis and drug resistance is complex, and not all drug resistance mechanisms can be attributed to apoptosis derangement.
- Table 1 should provide brief information on the role of individual listed miRNAS
Response: Thank you for this comment. There are many miRNAs listed in Table 1. The same miRNA may have more than one function. Table 1 has no enough space to provide the information of these miRNAs. Therefore, I have included the information for these miRNAs in the section “miRNA in apoptosis of colorectal cancer”. In this revision, I moved the part about 5-FU to the last two paragraphs such that the section can be more structured. In addition, more concepts of miRNA biogenesis are provided in this revision (lines 109-114, lines 116-120).
- Discussion could be renamed as Conclusion, and the Conclusion paragraph deleted as it is not related to the topic
Response: In the Discussion section, since the pathways related to CRC are discussed, these concepts may not be very suitable to be directly included in the Conclusion section. In addition, to address other reviewers’ comments, I include more concepts of pathways in the Discussion section. Therefore, I still keep the Discussion section. In the Conclusion section, I moved the miRNA part before the 5-FU part such that the first several sentences in the Conclusion are related to the miRNA topic. In addition, the sentence “Resistance to 5-FU is one of the main reasons for failure in the treatment” was removed.
- Language editing is needed.
Response: This revision was edited by MDPI English service.
Round 2
Reviewer 3 Report
The authors have improved the manuscript following the reviewers' instructions.